# Cellular Automata Modeling as a Tool in Corrosion Management

**DOI:** 10.3390/ma16176051

**Published:** 2023-09-03

**Authors:** Juan C. Reinoso-Burrows, Norman Toro, Marcelo Cortés-Carmona, Fabiola Pineda, Mauro Henriquez, Felipe M. Galleguillos Madrid

**Affiliations:** 1Centro de Desarrollo Energético de Antofagasta, Universidad de Antofagasta, Av. Universidad de Antofagasta 02800, Antofagasta 1271155, Chile; marcelo.cortes@uantof.cl (M.C.-C.); mauro.henriquez@uantof.cl (M.H.); 2Facultad de Ingeniería y Arquitectura, Universidad Arturo Prat, Av. Arturo Prat 2120, Iquique 1110939, Chile; notoro@unap.cl; 3Centro de Nanotecnología Aplicada, Facultad de Ciencias, Ingeniería y Tecnología, Universidad Mayor, Camino la Pirámide 5750, Santiago 8580745, Chile; fabiola.pineda@umayor.cl

**Keywords:** cellular automata, corrosion management, simulation corrosion, modeling corrosion

## Abstract

Cellular automata models have emerged as a valuable tool in corrosion management. This manuscript provides an overview of the application of cellular automata models in corrosion research, highlighting their benefits and contributions to understanding the complex nature of corrosion processes. Cellular automata models offer a computational approach to simulating corrosion behavior at the microscale, capturing the intricate interactions between electrochemical reactions, material properties, and environmental factors and generating a new vision of predictive maintenance. It reviews the key features of cellular automata, such as the grid-based representation of the material surface, the definition of state variables, and the rules governing cell-state transitions. The ability to model local interactions and emergent global behavior makes cellular automata particularly suitable for simulating corrosion processes. Finally, cellular automata models offer a powerful and versatile approach to studying corrosion processes, expanding models that can continue to enhance our understanding of corrosion and contribute to the development of effective corrosion prevention and control strategies.

## 1. Introduction

Corrosion refers to the irreversible reaction between a material and its environment, which can result in the degradation of the material and its properties. It is a natural process that occurs when a metal is exposed to oxygen and moisture [1]. It has a significant economic impact, representing approximately 3.4% of the global Gross Domestic Product (GDP) [2,3]. Moreover, corrosion can lead to disruptions in activities and decrease the efficiency and productivity of affected industries. While catastrophic failures due to corrosion have not been reported in solar thermal power plants, there are potential risks. For example, the Fukushima nuclear plant accident prompted the search for alloys that can withstand extreme operating conditions [4]. In the case of solar thermal power plants with thermal energy storage systems (TES), various corrosion mechanisms can occur, such as intergranular corrosion and mechanically assisted corrosion [5].

Cellular automata are computational models used to study the behavior of complex systems by simulating their evolution over time. Consist of a regular grid of cells, each of which can exist in a finite number of states and interact with its neighboring cells according to a set of rules. These rules govern the transition of cells from one state to another and can be based on simple local interactions or more complex global behaviors. The concept of cellular automata was first introduced by mathematician John von Neumann in 1940 [6], but cellular automaton pioneer John Conway popularized the idea in 1970. This manuscript recollects the main information about cellular automata models applied in corrosion management as an interesting tool for control and/or predictive maintenance of different types of corrosion and factors that contribute to corrosion.

## 2. Cellular Automata as Tools in Corrosion Management

Corrosion management refers to the systematic approach of preventing, controlling, and mitigating the effects of corrosion on materials and structures. It involves implementing strategies and practices to minimize the impact of corrosion, extend the service life, warrant safety, and reduce the maintenance and replacement of pieces or parts [7]. Effective corrosion management includes preventive measures such as using corrosion-resistant materials, applying protective coatings or inhibitors, and proper new designs and considerations for regular inspection and maintenance. Corrosion management entails the proactive implementation of preventive measures, including conducting regular inspections, performing thorough risk assessments, selecting appropriate materials, designing and constructing structures to minimize corrosion exposure, and employing a range of protective techniques, including coatings, corrosion inhibitors, cathodic protection systems, and other advanced technologies [8].

To effectively manage corrosion, a comprehensive understanding of the corrosion mechanisms associated with the Hydrogen Evolution Reaction (HER), Oxygen Reduction Reaction (ORR), and Oxygen Evolution Reaction (OER) is crucial, along with a deep knowledge of the operating environment and the factors that can either accelerate or mitigate corrosion. Additionally, it is essential to consider the applicable regulations and safety standards from NACE [8] or another standard that can apply regarding corrosion prevention across various industrial sectors, including petrochemicals [9], aeronautics [10], shipbuilding and maritime infrastructure [11,12,13,14], energy [15], mining [16], nuclear waste [17], microbial induced corrosion (MIC) [18], and atmospheric corrosion [19], among others. By incorporating this knowledge, industries can develop robust corrosion management strategies that ensure the safety, longevity, and optimal performance of their assets and infrastructure.

In the realm of corrosion management, the integration of computational techniques [20] and mathematical models [21] has emerged as an invaluable approach for comprehending and efficiently tackling the intricate behavior of corrosion processes. Among the array of techniques available, three stand out as particularly beneficial: cellular automata [22], Poisson processes [23], and Monte Carlo methods [24,25,26,27]. By harnessing these computational tools, researchers and engineers gain access to distinctive perspectives and capabilities, empowering them to simulate, analyze, and predict corrosion phenomena with remarkable precision and dependability.

The typical models, such as Cellular Automata, the Monte Carlo model, and Poisson processes, are mathematical models that can be utilized in different aspects of corrosion management. In the field of corrosion, cellular automata have proven valuable in providing information on the behavior of corrosion metals and alloys in contact with aggressive environments [28,29]. The cellular automata models generate interactions between electrochemical reactions, mechanical stresses, and material degradation, providing insights into the initiation, propagation, and evolution of various types of corrosion. By incorporating the influence of environmental factors, cellular automata models can be utilized to optimize corrosion-resistant materials.

Cellular automata models have proven to be valuable tools in corrosion management and mitigation strategies because it is possible to obtain predictive information on the behavior of the metal in contact with aggressive oxidizing agents [8,13,18,20,28,30]. They offer a computational framework for simulating and understanding the complex behavior of corrosion processes [31], enabling researchers and engineers to develop effective corrosion management techniques. The key applications of Cellular Automata models in corrosion management are (i) predicting corrosion propagation, which can simulate the propagation of corrosion in metallic structures such as pipelines, bridges, or storage tanks [32]. These models can predict the spatial and temporal evolution of corrosion damage and help in identifying critical areas with corrosion and planning targeted inspection and maintenance activities [33]. (ii) Cellular Automata help to understand the factors contributing to corrosion initiation, propagation, and the formation of localized corrosion features such as pits and crevices [34]. (iii) Cellular Automata can also be used to optimize corrosion management strategies by exploring different scenarios [22] and decision-making processes. These models can evaluate the cost-effectiveness of various corrosion control measures, optimize inspection and maintenance schedules, and assess the long-term performance and durability of materials under different environmental conditions. (iv) Cellular Automata: the assessment of corrosion-related risks in different scenarios, considering environmental parameters, material properties, and corrosion rules; these models can estimate the probability and severity of corrosion-induced failures, allowing for better risk mitigation and resource allocation [35]. (v) Cellular Automata can serve as an educational tool to enhance the understanding of corrosion processes among students and professionals in the field; by visualizing corrosion behavior and the effects of different factors [36], these models facilitate learning and provide a platform for exploring corrosion management strategies.

By utilizing Cellular Automata models in corrosion management, researchers and engineers can gain valuable insights into corrosion processes, optimize corrosion control measures, and develop proactive strategies to mitigate the detrimental effects of corrosion.

### 2.1. Fundamentals of Cellular Automata for Corrosion

The Cellular Automata model was initially proposed by John Von Neumann and corresponds to an idealization of a physical system where time and space are discrete and physical quantities take on a finite set of values [6]. Space is represented by a grid of cells, and each cell has a specific state. In a simplistic way, the state of a cell can be assigned as either alive or dead (0 or 1), indicating its characteristic.

To clarify, for an entity to be classified as a Cellular Automata, it must adhere to the following structure. A regular grid of cells that covers a part of a d-dimensional space.

A set 
$\Phi \left(\vec{r,}t\right)=\left\{\Phi_{1}\left(\vec{r,}t\right),\Phi_{2}\left(\vec{r,}t\right),\ldots ,\Phi_{m}\left(\vec{r,}t\right)\}\right.$
 of Boolean variables attached to each site r of the grid that gives the local state of each cell at time *t* = 0, 1, 2, …, n.

A rule *R* = {*R*_1_, *R*_2_, …, *R_m_*} that specifies the time evolution of the states 
$\Phi \left(\vec{r,}t\right)\;$
 in the following form:
(1)
$$\Phi_{j}\left(\vec{r,}t+1\right)=R_{j}\left(\Phi \left(\vec{r,}t\right),\;\Phi \left(\vec{r}+{\vec{\delta }}_{1},t\right),\Phi \left(\vec{r}+{\vec{\delta }}_{2},t\right),\ldots ,\;\Phi \left(\vec{r}+{\vec{\delta }}_{q},t\right)\right)$$

where 
$\vec{r}+{\vec{\delta }}_{k}$
 denotes cells belong to a specific neighborhood.

In the above definition, the rule *R* is applied to all sites simultaneously, leading to synchronous dynamics [36].

In the context of corrosion, the cell state can represent the presence of an element, a compound, or a corrosion product. The state of a cell changes over time depending on its own state and that of its nearest neighbors. Figure 1 shows a grid of cells where the cell undergoing a state change is represented in black, while the cells enclosed in a black frame (neighborhood) are the ones that influence this change, considering the state of their neighbors. This figure represents the three most commonly used types of neighborhoods: Von Neumann, Moore, and Margolus. In the Von Neumann neighborhood (a), four neighbors are considered, located to the east, north, west, and south of the changing cell. On the other hand, in the Moore neighborhood (b), there are eight neighbors surrounding the central cell. Finally, in the Margolus neighborhood, the change process is more complex as it depends on both space and time. Firstly, the space is divided into a 2 × 2 square. The first state change of cell *lr* occurs at an odd time, considering the neighbors in the upper square (*ul*, *ur*, *ll*, *lr*). In the next iteration, during even time, the state of cell *lr* is updated considering the neighbors in the square represented by dashed lines.

The neighborhood used traverses the entire grid, and through a set of transformation rules, the new state of each cell will be determined until the total update of the grid is completed, which represents the first generation (iteration) of the model. The rules and configuration of the Cellular Automata model in corrosion simulations are specific to each phenomenon to be studied.

Table 1 presents a comprehensive summary of the results published in popular science journals, which have been utilized to support our bibliographic review.

In addition to the cellular automata model, another computational approach used to simulate the phenomenon of corrosion is the finite element model (FEM). FEM is primarily based on the discretization of the domain into finite elements, allowing for a detailed representation of the geometry and properties of the metal. This is ideal for studying more localized behavior and investigating the microstructure of materials during corrosion. FEM utilizes differential equations of the physicochemical phenomena involved during corrosion, such as electrochemical equations, which are numerically solved to obtain solutions. FEM can study the corrosion process over time, considering environmental conditions and material responses.

On the other hand, the model presented in this review utilizes a representation based on the use of cell grids, providing a more macroscopic view of corrosion processes. The state of the cells is governed by predefined rules that represent the corrosion mechanism to be simulated, allowing for the visualization of emerging patterns and collective behaviors. These behaviors can be simulated over time by updating the states of the cells in successive iterations, capturing the progression of corrosion step by step.

Unlike FEM, which consumes more computational resources due to solving differential equations and domain discretization, the Cellular Automata model exhibits better efficiency in modeling larger and more complex systems.

Due to the demonstrated potential of the cellular automata model, this review primarily focuses on this model and does not consider others, such as FEM.

#### 2.1.1. Uniform Corrosion Model

Fairén et al. [31] analyzed the evolution of surface roughness in the studied corroded theoretical metal. They examined the agreement between a classical macroscopic description and a mesoscopic approach that accounts for the development of such roughness. They studied how morphology could influence the modeling, finding that the model could simulate the mesoscopic heterogeneity of the electrode surface and its impact on the uniform corrosion process. Part of their study determined that further research was still needed to fully understand the relationship between the electrochemical mechanism involved, the steps determining corrosion rate, and the morphology of the electrode surface.

Badiali et al. [37,38,39] studied the formation of films on a surface in the presence of corrosion, diffusion, and precipitation at the growth front. Obtaining results that show that the growth of the layer follows a parabolic law and that the model can be useful for predicting and controlling corrosion growth in metallic structures, pipes, and equipment, as well as for developing new corrosion-resistant materials. Chen et al. [40,41] studied the corrosion and oxidation mechanisms of stainless steel in lead-bismuth eutectic (LBE) environments, a liquid metal used as a coolant in some advanced nuclear reactors and accelerator-driven systems; in [42], the medium was supercritical water. The model considered diffusion, reaction, and precipitation processes occurring on the steel surface, and a mesoscopic description was used to explore the general characteristics of the evolution of the involved processes. Additionally, they aimed to predict how oxide scale removal occurs and how it affects metal corrosion, concluding that the Cellular Automata model is useful for understanding these mechanisms. Chen et al. [43] focused on simulating the growth of the oxide layer in chromium-containing stainless steels. Their objective was to verify whether the stochastic nature of the Cellular Automata model for uniform corrosion would generate unstable or unreasonable results deviating from the deterministic model for uniform corrosion of steel in flowing LBE. Their results showed that the Cellular Automata model is stable and reliable for simulating the thickness of the oxide layer. Chen et al. [35] proposed a numerical simulation method to predict the evolution of uniform corrosion damage to the outer steel tube in concrete-filled tubular columns subjected to corrosive environments. They discussed the influence of solution concentration and the probability of dissolution of corrosive agents. The obtained results were compared with theoretical solutions and experimental findings. They concluded that different concentrations of corrosive agents have different impacts on the degree of corrosion damage. Ren et al. [44] simulated the uniform corrosion of aluminum in various environmental conditions. They defined corrosion rules based on the actual electrochemical reactions and examined the corrosion process at a mesoscopic scale. By studying corrosion formation and modifying the rules for different concentrations of corrosive solutions and ambient temperatures, they gained insights into corrosion mechanisms. The simulation results contribute to a better understanding of corrosion and can aid in its prevention and mitigation, especially in aeronautical structures where corrosion can lead to fatigue-induced damage, compromising structural integrity. Li et al. [45] sought to gain a better understanding of the corrosion evolution of marine structural steel in the tidal zone and how corrosion can be prevented or mitigated in this environment. Their objective was to comprehend the corrosion mechanisms in the tidal zone, identify the factors influencing the corrosion rate, and propose possible strategies to prevent or mitigate corrosion in structures and equipment used in marine environments, ultimately offering solutions to extend their lifespan. Wang et al. [32,46] investigated the corrosion of an Inconel alloy 625 and Hastelloy X (Excellent oxidation resistance up to 1200 °C) in contact with molten chloride salt and explored methods to enhance its corrosion resistance. A simplified model was established based on the reactive diffusion of corrosive gas and the metallic substrate. Simulations were conducted using the Cellular Automata method.

The physical model from the experimental work determined that the main reactions involved are:
(2)
$$\mathrm{Cr}+2{\mathrm{MgCl}}_{2}+O_{2}\;\to {\mathrm{CrCl}}_{4}+2\mathrm{MgO}$$


(3)
$$\mathrm{Ni}+1/2O_{2}\to \mathrm{NiO}$$


(4)
$$2{\mathrm{CrCl}}_{4}+2\mathrm{MgO}+O_{2}\to {\mathrm{MgCr}}_{2}O_{4}+3{\mathrm{Cl}}_{2}+{\mathrm{MgCl}}_{2}$$


(5)
$$2{\mathrm{Cl}}_{2}+\mathrm{Cr}\to {\mathrm{CrCl}}_{4}$$


They assumed the presence of corrosive substances in the molten salt from the beginning, where O_2_ and H_2_O diffused into the molten salt from the air, leading to the formation of HCl and Cl_2_ through a set of reactions. The dissolved O_2_ in the salt rapidly reacts with Cr (2). When the Cr content is insufficient, O_2_ reacts with Ni (3). The chloride containing Cr (mainly CrCl_4_) forms a protective spinel layer composed of MgCr_2_O_4_ through (4). On the other hand, Cl_2_ can react with Cr, resulting in a chromium-depleted layer (5).

To establish the Cellular Automata model, the chemical reactions involved in the corrosion process were simplified by assigning letters to the compounds or elements present. A = Cr; B = Ni; O = O_2_; C = Cl_2_; D = CrCl_4_; Mg = MgO; P = MgCr_2_O_4_; BO = NiO.

The lattice sites were classified into fixed compounds (A, B, Mg, P, and BO) and mobile compounds (O, C, and D). The interaction of mobile compounds within the lattice occurs through a probabilistic random walk process. As an example, Figure 2 shows four grid schematics of the Cellular Automata model. In Figure 2a, a grid is depicted with the fixed compounds located on it. In Figure 2b, the location of mobile compounds that diffuse toward the metal is shown. In Figure 2c, the elements that can diffuse in the outer corrosion layer are present, and in Figure 2d, a lattice is shown where all compounds present in the process are located.

In Equations (6)–(9), the modified chemical reaction and transformation rules for the Cellular Automata model are shown. The author does not consider the consumption and formation of MgCl_2_ due to the involvement of O_2_ dissolution in chemical reactions (2) and (4).

(6)
O + A → D + Mg


(7)
O + B → BO


(8)
D + Mg + O → P + C


(9)
C + A → D


The Margolus neighborhood was used to induce state changes in the different cells across generations. Figure 3 displays the results obtained by the model, where the left side (black color) represents the molten salt and the right side (purple color) represents the metal under study. As the iterations progress, the growth of the oxide layer is observed. The number of iterations can be interpreted as representing an experimental time scale.

A comparison is shown in Figure 4 between the simulation results and a scanning electron microscopy (SEM) image. It was determined that 64,000 iterations correlate with the experimental results from 21 days of exposure, demonstrating that the Cellular Automata simulation yields satisfactory results in the study of high-temperature corrosion of Fe-Cr alloys in the presence of molten salts.

#### 2.1.2. Localized and Pitting Corrosion Model

Localized corrosion encompasses corrosion processes that occur in specific areas of a material, leading to localized damage [47]. Cellular automata simulate the initiation and propagation of localized corrosion, providing insights into factors influencing its occurrence, growth patterns, and evolution over time.

Di Caprio et al. [22,48] presented an electrochemical model for the corrosion of metals in contact with liquids based on the description of chemical and electrochemical reactions occurring at the metal-liquid interface, which was simulated using Cellular Automata. They investigated the corrosion process at each metallic site at the interface and compared the simulation results with experimental data. The proposed model reproduced experimental facts and trends without an explicit separation between anodic and cathodic sites. Overall, the study demonstrated that cellular automata-based models are a useful tool for simulating complex systems such as metal corrosion and can be adapted to include different rules and conditions according to the specific needs of the system under study. Cheng et al. [49] simulated the growth of metastable corrosion pits. The objective was to gain a better understanding of the mechanism behind the growth of these pits and compare the simulated results with experimental data. The researchers established a relationship between the current and current density of the pit over time to illustrate the mechanism of metastable pit growth. Additionally, they developed an optimal range of parameters for the simulation that allowed for visualization of the complete process of pit growth, including its geometry. The study built on previous work on the mechanism and electrochemical and mass transfer steps associated with the pitting corrosion process. Wang et al. [50] reproduced the interactions between metastable pits in stainless steel and analyzed how different factors affect their growth and stability. The model included corrosion, passivation, salt film hydrolysis, and hydrogen ion diffusion. Based on the model, they concluded that it is capable of accurately simulating the interactions between metastable corrosion pits in stainless steel. Wang et al. [51] investigated the interaction of metastable corrosion pits in stainless steel under mechano-electrochemical effects using an updated cellular automaton/finite element model, elucidating the mechanisms for pit interactions. In the study, they considered an electrochemical system where stainless steel is exposed to an aggressive chloride solution. The stainless–steel samples had passive layers of surface oxide, which can be damaged in the presence of chloride anions. Two breakdown locations in the passive film were included to study only the pit propagation and not its nucleation. They concluded that the cellular automaton/finite element model used was effective in predicting the interaction of corrosion pits on a mesoscopic scale. Torska et al. [52] discussed the fracture dynamics in the investigated areas and performed a comparative analysis of pitting corrosion rates under real and simulated conditions. They demonstrated that the simulation procedure using cellular automata accurately reproduced the physics of corrosion. The study proposed a new modeling algorithm and local transition rules for the automaton cells used in simulating pitting corrosion images. Hu et al. [53] simulated pitting corrosion in nickel alloys. The authors employed four fundamental elements of the model to simulate electrochemical reactions, chemical reactions, and diffusion processes. The obtained results were qualitatively and quantitatively compared with experimental data and analogous findings cited in the literature. The study suggested that this type of model could be useful in gaining a better understanding of corrosion processes and developing corrosion-resistant materials for various industrial applications. Fatoba et al. [54] employed a combination of cellular automata and finite element analysis to simulate the growth of localized corrosion under the influence of applied stress. The results demonstrated that mechanical effects, such as plastic deformation, accelerated the rate of localized corrosion development. The study focused on low-alloy steel, but the findings may apply to other materials as well. Rujin et al. [55] established a mathematical model based on a statistical approach to describe the evolution of pitting corrosion. Additionally, they proposed a 3D stochastic Cellular Automata model to replicate the simultaneous initiation and growth processes of pits. The findings of the study can contribute to a better understanding of the laws governing pitting corrosion evolution and provide valuable insights into its prevention and control. To simulate localized corrosion, Pérez–Brokate et al. [34,56] aimed to gain a better understanding of hidden corrosion processes and pitting corrosion within corrosion cells by using a stochastic Cellular Automata model. They studied the morphology, propagation, and influence of coupled diffusion within the corrosion cavity.

In the physical-chemical model, simplifications were made by assuming simplified electrochemical and chemical reactions (excluding contaminants and considering only H^+^ and OH^−^). Localized corrosion, being a multi-scale phenomenon, depends not only on atomic-scale surface phenomena but also on macroscopic environmental conditions.

The electrochemical reactions used:

Anodic reactions.

(10)
$$M+H_{2}O\;\to {\mathrm{MOH}}_{\mathrm{aq}}+H^{+}+e^{-}$$


(11)
$$M+{\mathrm{OH}}^{-}\;\to {\mathrm{MOH}}_{\mathrm{solid}}+e^{-}$$


Cathodic reactions.

(12)
$$H^{+}+{\;e}^{-}\to \;\frac{1}{2}H_{2}$$


(13)
$$H_{2}O\;+{\;e}^{-}\to \;\frac{1}{2}H_{2}+{\mathrm{OH}}^{-}$$


Authors call these Spatially Joint (SJ) reactions when the anodic and cathodic half reactions occur at the same location:
(14)
$$M+H_{2}O\;\to {\mathrm{MOH}}_{\mathrm{aq}}+\frac{1}{2}H_{2}$$


(15)
$$M+H_{2}O\;\to {\;\mathrm{MOH}}_{\mathrm{solid}}+\frac{1}{2}H_{2}$$


(16)
$${\mathrm{MOH}}_{\mathrm{solid}}\to {\mathrm{MOH}}_{\mathrm{aq}}$$


The anodic and cathodic sites are electrically connected through the metal and the solution. When they are separated, they are referred to as Spatially Separated Electrochemical (SSE) reactions. The diffusion of SSE reactions results in the generation of H^+^ and OH^−^ ions, mimicking a random walk. When these ions interact, neutralization takes place, as depicted in the study through the following reaction:
(17)
$$H^{+}+{\mathrm{OH}}^{-}\to H_{2}O$$


The 3D lattice representation is depicted in Figure 5, illustrating a passivated metal in contact with a neutral electrolyte, and the cathodic reaction takes place at a random point (Figure 5a). The H^+^ and OH^−^ ions generated in the first reaction diffuse in the electrolyte. Each lattice site was determined by the dominant species. In Figure 5b, the solid sites represent the metal (M), the metal in contact with the electrolyte (R), and the oxide layer providing passivity to the metal (P). Additionally, the electrolyte sites were differentiated into three different states based on pH. An acidic site (A), a basic site (B), and a neutral site (E) were considered. The Moore neighborhood was employed for the simulation.

Figure 6 shows the evolution of the pit. Perez−Brokate et al. indicated the presence of an anodic half−reaction at the initial defect of the passive layer. The ions are dispersed randomly in the electrolyte, resulting in the growth of the pit when an H^+^ ion encounters the metal, leading to an electrochemical reaction. As the pit reaches a certain size, the concentration of acidic ions increases, causing instability and further enlargement of the pit. Each state of the network has a different color. Cells in green correspond to H^+^, in blue to OH^−^, in red to the reactive site, and in magenta to the passive site. Cells of neutral solution were not represented to enhance visibility.

The model was considered a valuable tool for gaining a better understanding of localized and pitting corrosion processes and developing more effective strategies for corrosion prevention and treatment.

#### 2.1.3. Stress Corrosion Cracking (SCC) Model

Stress corrosion cracking is a phenomenon where the combination of tensile stress and a corrosive environment leads to crack initiation and propagation in a material [57]. Cellular automata model the complex interactions between mechanical stress, electrochemical processes, and material degradation, helping to understand the conditions under which SCC occurs and its progression.

Zhu et al. [58] focused on predicting the service life of concrete bridges and preventing chloride-induced corrosion. They utilized 3D Cellular Automata, which accurately simulate chloride diffusion in concrete. Multiple factors were considered, such as ambient relative humidity, temperature variations, stress, water−cement ratio (*w*/*c*), concrete degradation, corrosion propagation, cracks, and time. The study provides a useful tool for predicting the service life of the bridge and designing preventive measures against chloride-induced corrosion in concrete bridges. Liu et al. [59] modeled the SCC process in steel pipes. They employed a combination of finite element analysis and cellular automata techniques to simulate the initiation and propagation of cracks.

The electrochemical model used considered only the species H_2_O, Fe, Fe^2+^, H^+^, and FeOH^+^. The rules for corrosion evolution and cell diffusion were expressed in the following equations:
(18)
$$\mathrm{Fe}\to {\mathrm{Fe}}^{2+}+2e^{-}$$


(19)
$${\mathrm{Fe}}^{2+}+2H_{2}O\;\to {\mathrm{FeOH}}^{+}+2H^{+}$$


Based on the equations above, the authors formulated the rules of the Cellular Automata model, where the diagram in Figure 7 represents the spatial layout of the model. Sites W represented a non-corrosive neutral solution (H_2_O), H represented a corrosive acidic solution (H^+^), and site M represented the metal, which dissolves to form site R after being in contact with the corrosive solution. The site R represents the active metal (Fe^2+^), and the site P represents the corrosion product (FeOH^+^).

The evolution rules for corrosion in the Cellular Automata model were formulated based on the previous Equations (18) and (19). Figure 8 shows the rules for the oxidation of Fe and the hydrolysis of Fe^2+^ to FeOH^+^. When a site M is in contact with at least one site H, reaction (18) will occur with a corrosion probability P_Corr, transitioning from site M to site R. Additionally, when a site R is in contact with at least two sites W, Fe^2+^ hydrolyzes to FeOH^+^ with a probability P_Hyd, and both site R and the two W sites are replaced by the corrosion sites P and two H sites, respectively.

The results showed in Figure 9 demonstrated that before crack initiation, pitting corrosion was controlled by anodic reactions and mechanical factors, with electrochemical corrosion playing a significant role. During the crack propagation process, the mechano−chemical effects induced by plastic deformation promoted anodic dissolution at the crack tip, driving its propagation. This study may have significant implications for the pipe industry and help develop more effective strategies for corrosion prevention.

#### 2.1.4. Intergranular Corrosion Model

Intergranular corrosion occurs along the grain boundaries of a material, typically due to variations in composition or microstructure. Cellular automata can simulate the interactions between different grain boundaries, their susceptibility to corrosion, and the role of environmental factors in intergranular corrosion [60].

Chen et al. [61] developed a cellular automaton model to predict the susceptibility to intergranular corrosion in austenitic stainless steel and provide useful information on how to prevent or mitigate this problem. The results demonstrated how material properties change under different heat treatment and sensitization conditions. Additionally, they investigated factors affecting intergranular corrosion, such as the evolution of chromium-rich carbide precipitation and chromium concentration distribution [62]. They concluded that the Cellular Automata model can be valuable for improving the design and performance of materials used in industrial applications.

Lishchuk et al. [63] utilized cellular automata to describe the propagation of intergranular corrosion and made simplifying assumptions that enabled them to predict the corrosion rate. The results of the model demonstrated good qualitative and quantitative agreement with experimental data regarding the advancement of the corrosion front.

Jahns et al. [64,65] developed a simulation to predict internal corrosion during high-temperature applications in metal alloys. They utilized the model to describe diffusion-controlled precipitation processes and enhance our understanding of high-temperature corrosion in metal alloys. The researchers’ goal was to develop more effective strategies to prevent or mitigate corrosion.

Di Caprio et al. [66,67,68,69] conducted various studies on intergranular corrosion using 2D and 3D Cellular Automata models. They aimed to predict the rate and pattern of corrosion in stainless steel exposed to corrosive solutions. They quantitatively analyzed the surface morphology of the steel and grain boundary structures. They presented Cellular Automata methodologies to understand and prevent corrosion, and experimental validation yielded positive results.

Wang et al. [70] focused on investigating the corrosion behavior of a nickel-based alloy in a molten chloride salt mixture. The Cellular Automata model enabled them to reproduce the distribution of corrosion products and components as well as changes in the morphology and thickness of the corrosion layer over time. The results of the study can help predict the corrosion behavior of different metals in similar molten salt environments.

The mechanism of high-temperature corrosion is relatively complex. In the study conducted by Wang et al. [46], due to the alloy being primarily composed of Cr, Fe, and Ni elements, which play a significant role in corrosion resistance and alloy microstructure, only Cr, Fe, and Ni were considered in the model, excluding other elements present in lower concentrations.

The mechanism considered by the authors primarily involved the migration of metallic elements, substance consumption, and generation of corrosion products through the following equations:
(20)
$$4\mathrm{Cr}+3O_{2}\to 2{\mathrm{Cr}}_{2}O_{3}$$


(21)
$$4\mathrm{Fe}+3O_{2}\to 2{\mathrm{Fe}}_{2}O_{3}$$


(22)
$$2\mathrm{Ni}+O_{2}\to 2\mathrm{NiO}$$


(23)
$$\mathrm{Cr}+2{\mathrm{Cl}}_{2}\to {\mathrm{CrCl}}_{4}$$


(24)
$$2\mathrm{Fe}+3{\mathrm{Cl}}_{2}\to 2{\mathrm{FeCl}}_{3}$$


(25)
$$2{\mathrm{CrCl}}_{4}+3O_{2}\to 2{\mathrm{Cr}}_{2}O_{3}+4{\mathrm{Cl}}_{2}$$


(26)
$$4{\mathrm{FeCl}}_{3}+3O_{2}\to 2{\mathrm{Fe}}_{2}O_{3}+6{\mathrm{Cl}}_{2}$$


Like the studies presented in the uniform corrosion section, the physical model was simplified to facilitate the programming language.

(27)
EE+O→YY


(28)
DD+O→Y


(29)
AA+O→AO


(30)
EE+c→D


(31)
DD+c→FF


(32)
D+O→YY+c


(33)
FF+O→Y+c

where: AA = Ni, DD = Fe, EE = Cr, O = O_2_, c = Cl_2_, Y = Fe_2_O_3_, YY = Cr_2_O_3_, AO = NiO, D = CrCl_4_, and FF = FeCl_3_. All elements except for O, c, and D were considered fixed in the lattice. The positions of O and c were randomly predetermined with an assigned probability, and the sites AA, DD, and EE represent the alloy elements.

To consider the simulation of intergranular corrosion, the authors evaluated the effect of grain size. To obtain the initial structure, they simulated the grain growth process at different time steps, as shown in Figure 10.

Therefore, their simulation of intergranular corrosion consisted of obtaining the initial microstructure model, which was combined with the Cellular Automata model. The result of the model is shown in Figure 11, which displays different structures of the model at various iterations. The growth of intergranular corrosion in the studied steel is observed.

## 3. Modeling Approach and Methodology for Simulating Corrosion Phenomena

Methodologies play a crucial role in establishing best practices, enhancing efficiency, generating evaluation criteria, and promoting result reproducibility by providing an organized structure. This review focuses on the predominant methodology observed in the studies, which effectively facilitates the structured organization of the modeling process using the Cellular Automata model.

The methodology for simulating corrosion phenomena involves several key steps and considerations. These are presented and detailed in the flowchart in Figure 12.

**Table 1 materials-16-06051-t001:** Summary of the results published.

Cellular Automata Type	Neighborhood	Model Type	Boundary	Lattice Size	Rules	Cycles	Corrosion Type	Electrolyte	Material	Temperature (°C)	Exposure Time (h)	Validation Type	Ref.
LGA	n−vector	Probabilistic	Periodic	-	-	-	-	-	-	-	-	Boltzmann hypothesis	[71]
2D	n−vector	Probabilistic	Periodic	200 × 160	3	48,000	Kinetic of internal oxide precipitation	Oxygen	Theoretical Metal	-	-	Theory of phase equilibrium	[72]
Theoretical	Moore	Probabilistic	Periodic	>1000 × 1000	8	-	Uniform	Theoretical Electrolyte	Theoretical Metal	-	-	Previous work data	[31]
2D	Von Neumann	Probabilistic	Periodic	2000 × 1000	2	10,000 to 90,000	Uniform	Theoretical Electrolyte	Theoretical Metal	-	-	-	[37]
2D	Von Neumann	Probabilistic	Periodic	1000 × 1000	3	200,000	Uniform	Theoretical Ionic Solution	Theoretical Metal	-	-	Parabolic law	[38]
2D	Von Neumann	Deterministic	Periodic	600 × 20,000	2	150	Uniform	Theoretical Electrolyte	Theoretical Metal	-	-	Mott and Cabrera Parabolic’s law	[39]
2D	Von Neumann	Probabilistic	Periodic	1000 × 1000	2	25,000	Uniform	Lead−bismuth Eutectic	Stainless Steel	550	3000	Wagner theory	[40]
3D	-	-	-	100 × 100	-	-	Intergranular	Sulfuric Acid	AISI 304 and AISI 316 SS	1100	0−400	Electrochemical potentiodynamic reactivation	[61]
2D	Moore	Probabilistic	-	500 × 500	2	200,000	Uniform	Lead−bismuth Eutectic	Stainless Steel (Fe, Cr)	535	3000	Wagner theory	[41]
LGA-2D	Von Neumann	Probabilistic	-	2000 × Ly	3	-	Passivation	Theoretical Electrolyte	Theoretical Metal	-	-	Passivation theory	[73]
2D	Moore	Deterministic	-	200 × 200	2	3	Intergranular	Sensitization Treatments	SS 316	1100	2	-	[62]
2D	Von Neumann	Probabilistic	Periodic	-	2	7,200,000	Localized	Theoretical Electrolyte	Theoretical Metal	360	-	Experimental	[48]
2D	Von Neumann	Probabilistic	-	1000 × 1000		399,577	Localized	Theoretical Electrolyte	Theoretical Metal	-	-	Pistorius experimental results	[49]
2D-3D	Moore	Probabilistic	Periodic	640 × 320–240 × 280 × 240	3	-	Intergranular	Chloride Solutions	AA2024 Alloy	-	144	Eperimental	[63]
LGA-2D	Von Neumann	Probabilistic	-	2000 × Ly	2	-	Passivation	Theoretical Electrolyte	Theoretical Metal	-	-	Passivation theory	[74]
2D	Moore	Probabilistic	-	500 × 500	2	5000	Uniform	Supercritical Water	Inconel 625	600	1000	Experimental	[42]
2D	Von Neumann	Probabilistic	Periodic	900 × 300	2	70,000	Uniform, localized	Theoretical Electrolyte	Theoretical Metal	300–360	-	Experimental	[22]
2D	Von Neumann and Moore	Probabilistic	Periodic	1000 × 1000	14	7679	Localized, passivation	Theoretical Electrolyte	Theoretical Metal	-	-	Theoretical corrosion and passivity phenomena	[75]
2D	Von Neumann and Moore	Probabilistic	Periodic	1000 × 1000	11	4000	Crevice, passivation	Theoretical Electrolyte	Theoretical Metal	-	-	Experimental	[76]
2D	Moore	Probabilistic	-	500 × 500	2	200,000	Uniform	Lead−bismuth Eutectic	Stainless Steel	-	-	Chi-square of goodness−of−fit	[43]
2D	Von Neumann and Moore	Probabilistic	-	512 × 512	3	2000	Oxidation and nit	Nitrogen	Ni−20Cr−2Ti Alloy	1100	100	Experimental data	[77]
3D	Moore	Probabilistic	Periodic	151 × 200 × 200	3	200	Pitting	Hydrogen Carbonate Solution	AISI 1040	-	0.5	Experimental data	[78]
2D	Von Neumann and Moore	Probabilistic	-	512 × 512	3	20,000	Oxidation and nitridation	Nitrogen	Ni−20Cr−2Ti Alloy	1100	100	Experimental data	[79]
3D	Moore	Probabilistic	Periodic	-	4	124	Pitting	Concrete Pore Solution	Q235 Carbon Steel	-	480	Experimantal data	[80]
2D	Moore and Von Neumann 2nd order	Probabilistic	-	1000 × 1000	4	1000	Pitting	Theoretical Solution	Theoretical Metal	-	-	Experimental data	[81]
2D	Von Neumann	Probabilistic	-	1024 × 1024	2	-	Pitting	Neutral Solution	Stainless Steel	-	-	Experimental data	[50]
2D	Moore	Deterministic	Reflect	-	2	-	Pitting	Theoretical Solution	Theoretical Metal	-	-	Experimental data	[82]
2D	Von Neumann	Probabilistic	Periodic	1024 × 1024	5	-	Pitting	Chloride Solution	Stainless Steel	-	-	Experimental data	[51]
2D	Von Neumann 2nd order	Probabilistic	-	-	3	-	Pitting	Ferrous Chloride	D16T Alloy	Room	-	Experimental data	[52]
2D	Von Neumann	Probabilistic	Periodic	2000 × 2000	3	6000	Pitting	Water	Alloy 690	-	1600	Experimental data	[53]
2D	Von Neumann and Moore	Probabilistic	-	500 × 500	3	3000	Intergranular	Air	Ni−Cr20−2Ti−Mn Steel and Inconel 625Si	1100	100	Experimental data	[64]
3D	Moore	Probabilistic	-	512 × 512 × 512	4	>5000	Occluded, localized	Theoretical Acid−base Solution	Theoretical Metal	-	-	Experimental data	[56]
3D	-	Probabilistic	Periodic	512 × 512 × 4096	-	4000	Intragranular	Acid Solution	Stainless Steel	-	7000	Experimental data	[66]
2D	Von Neumann	Probabilistic	-	-	3	25,000	Electrochemical oxidation	HClO_4_ Aqueous Solution	Ketjenblack ES DJ 600	600	30	Experimental data	[83]
2D	Von Neumann and Moore	Probabilistic	-	512 × 512	3	-	Intergranular oxidation	Atmospheric Air	Alloy 80a and X60 Steel	1100	100	Experimental data	[65]
2D	Moore and Von Neumann extend	Probabilistic	Periodic	1000 × 1000	-	2000	Uniform	Acid Rain	Concrete Filled Square Steel Tubular Columns	-	-	Theoretical and experimental	[35]
3D	Moore	Probabilistic	-	256 × 256 × 256	7	45,000	Pitting	Theoretical Acid−base Solution	Theoretical Metal	-	-	Experimental data	[34]
3D	Moore	Probabilistic	Periodic	512 × 512 × 512	7	7000	Generalized	Theoretical acid−base Solution	Theoretical Metal	-	-	Experimental data	[84]
CA-FE-2D	Moore	Probabilistic	-	2000 × 1000	4	-	Localized	NaCl	X65 Steel	-	-	Experimental data	[54]
3D	Von Neumann	Probabilistic	Constant and periodic	100 × 100 × 100	4	400	Atmospheric corrosion	Atmospheric Air	Wheatering Steel	-	-	Theoretical and experimental data	[85]
3D	Von Neumann	Deterministic	Periodic	100 × 100 × 100	5	1000	Uniform	Corrosion Solution	Aluminum Alloy	20	-	-	[44]
3D	Von Neumann	Probabilistic	-	7500 × 7500 × 1500	-	-	Pitting	Salt−Spray	Q345 Steel	35	1440	Experimental data	[55]
2D	Margolus	Probabilistic	Rigid	400 × 400	4	25,000	Uniform	Chloride Molten Salt	Inconel625 Alloy	600	504	Experimental data	[32]
LGA-3D	Von Neumann	Probabilistic	Periodic	-	-	-	Passivation	Electrolyte Solution	Metal Electrode	-	-	Passivation theory	[86]
3D	12−neighbors	Probabilistic	Periodic	1280 × 1280 × 1280	-	-	Intergranular	Solution	Stainless Steel	-	2000	Experimental data	[67]
2D-bi plane	Von Neumann	Probabilistic	-	500 × 500	5	100	Pitting	Solution	Steel Wires	-	-	Experimental data	[87]
3D	-	-	-	2048 × 2048 × 256	3	350	Intergranular	-	Stainless Steel	-	-	Experimental data	[68]
2D	Margolus	Probabilistic	Periodic	400 × 400	7	70,000	Uniform, intergranular	NaCl−CaCl_2_ Molten Salt	Hastelloy X	600	504	Experimental data	[46]
3D	Margolus	Probabilistic	Periodic	1280 × 1280 × 1280	-	21,555	Intergranular	Nitric Acid Solution	AISI 3010L Stainless Steel	111	17,783	Experimental data	[69]
2D-3D	Margolus	Probabilistic	Periodic	40 × 400–100 × 100 × 100	10	50,000	Intergranular, pitting	NaCl−KCl−ZnCl_2_ Molten Salt	Inconel 625 Allow	700	168	Experimental data	[70]
3D	Von Neumann	Probabilistic	Periodic	1000 × 1000 × 1000	5	-	Uniform, pitting	Sea Water Solution	Q235 Steel		168	Experimental data	[45]
2D	Moore	Probabilistic	-	1024 × 1024 × 128	4	2,000,000	Aqueous	Theoretical Acid−base Solution	Theoretical Metal	50	853	Experimental data	[88]
2D	Von Neumann	Probabilistic	-	250 × 250	2	2000	Cracking	Electrolyte Solution	Carbon Steel	27	-	Experimental data	[59]
3D	Von Neumann	Probabilistic	-	2000 × 2000 × 1000	2	-	Cracking	Chloride Solution	Steel bar	23	672	Experimental data	[58]
3D	Moore	Probabilistic	Periodic	104 × 104 × 104	5	-	Atmospheric corrosion	Atmospheric Air	Structural Steel	-	-	Experimental data	[33]

## 4. Future Perspectives and Emerging Trends

The development of Cellular Automata models applied to the phenomenon of corrosion has been observed throughout history. Initially, studies focused primarily on simulating the different mechanisms involved in corrosion. For example, as early as 1997, the random walk of oxide precipitation was explored. In 2004, a simple corrosion mechanism was theoretically simulated by incorporating the anionic and cationic equations associated with the corrosion process. Concurrently, in 2004, efforts were made to expand the application of cellular automaton models by simulating the corrosion and passivation phenomena of a metal without specifying a particular metal but rather seeking to mimic the formation of a particle film representing the passivating products based on theoretical grounds. Since 2008, studies on localized pitting corrosion growth and the corrosion of steels in specific environments, such as the oxidation of Inconel 625 in supercritical water, have emerged. This advancement has expanded the knowledge and application of the cellular automaton model, leading to recent simulations of corrosion processes at high temperatures. For instance, in 2022, the corrosion process of a Ni-based alloy in molten chloride salt was successfully simulated.

It can be observed that the Cellular Automata model has a similar structure in many of the reviewed studies. The starting point is a known theoretical or experimental phenomenon that is to be simulated using Cellular Automata. Next, the physicochemical or electrochemical model governing the process is defined and translated into a simplified language that facilitates programming. Each study establishes its own reaction and transformation rules, which depend on the initially proposed physicochemical or electrochemical model to simulate the desired process.

The summary of the table shows that, due to the nature of the simulated phenomenon, almost all studies utilize a probabilistic cellular automaton model, except in specific cases. This is because corrosion occurs spontaneously and randomly on the metal surface. In addition, the choice of neighborhood type is predominantly divided between the two most popular ones: the Moore and Von Neumann neighborhoods. Margolus neighborhood is employed in cases where authors indicate its suitability based on the required computational resources. It is not possible to determine a trend regarding the size of the grid used, as each study requires specific dimensions. Similarly, it is not possible to do so with the number of reaction and transformation rules utilized, for the same reason. However, a trend can be observed in the choice of boundary conditions made by the authors. The use of periodic boundary conditions seems ideal for simulating corrosion processes, as it allows for the simulation of large-scale systems using a reduced section of space.

Cellular automata models have emerged as powerful tools for analyzing various types of corrosion, offering valuable insights into their complex behavior [67,69,70]. These models capture spatial and temporal variations, enabling a thorough examination of corrosion phenomena. Notably, cellular automata models have successfully analyzed specific corrosion types such as stress corrosion cracking [59], pitting corrosion [34], marine corrosion [45], and more. Through these simulations, critical areas prone to corrosion can be identified, corrosion control measures can be evaluated, and corrosion-induced failures can be predicted, providing valuable perspectives for effective corrosion management [20]. In this review, we focus on the Cellular Automata model as opposed to similar models such as Finite Element or Monte Carlo models. This is done to delve deeper into specific aspects, the key characteristics, and the applications of the Cellular Automata model in the context of corrosion. Additionally, unlike the Finite Element model, for example, the Cellular Automata model, with its simple rules, can be more efficient in simulating large and complex systems. It provides a more macroscopic representation of the corrosion process, allowing for the visualization of patterns and behaviors over time. This offers a broader and more generalized perspective on the phenomenon of corrosion, which may be of interest to a wider audience.

When developing cellular automata models to simulate corrosion phenomena, it is crucial to consider a range of environmental parameters that have a substantial impact on corrosion performance. These parameters reflect the realistic conditions under which corrosion takes place [85]. Firstly, the composition of the electrolyte solution surrounding the material holds the utmost importance in determining the corrosive environment. Factors such as pH, chloride concentration, dissolved oxygen levels, temperature, and other relevant chemical species all contribute to this environment [32]. Secondly, the movement of the electrolyte solution, also known as flow or mass transport, plays a significant role in corrosion processes. It is imperative to account for these parameters to create accurate and comprehensive cellular automata models for studying corrosion behavior [65]. Consideration of flow patterns, velocity, and diffusion rates within cellular automata models is essential to simulating the effects of convection, mass transfer, and concentration gradients on corrosion behavior [64]. Thirdly, these models should account for the specific environmental exposure conditions relevant to the corrosion scenario, such as marine environments, industrial atmospheres, underground conditions, or exposure to specific chemicals [35]. The inclusion of temperature as a critical environmental parameter is crucial, as thermal effects can impact corrosion rates by influencing electrochemical reaction kinetics, diffusion rates, and material properties. By incorporating temperature variations into the cellular automata model, a more comprehensive understanding of corrosion behavior under different thermal conditions can be achieved [41,64].

In some corrosion scenarios, such as atmospheric corrosion, humidity and moisture levels play significant roles [85]. A well-designed cellular automata model enables the simulation of localized corrosion phenomena related to moisture accumulation or the presence of electrolyte films [48]. Additionally, it is important to consider the influence of ultraviolet (UV) radiation from sunlight, especially in outdoor environments. UV radiation can induce photochemical reactions, affect surface properties, and alter the electrochemical behavior of materials [89]. Therefore, incorporating the effects of UV radiation into the cellular automata model provides a more accurate representation of corrosion behavior. By considering these various environmental parameters and incorporating them into the cellular automata models, a comprehensive and realistic simulation of corrosion behavior can be achieved, facilitating a deeper understanding of the corrosion processes and aiding in the development of effective corrosion mitigation strategies. Through the evaluation of corrosion mechanisms, the development of corrosion prevention strategies, and the assessment of material performance in specific environments by cellular automata models, researchers can obtain a more comprehensive understanding of corrosion processes and accurately simulate the behavior of materials under realistic conditions [63]. The integration of data-driven approaches, such as machine learning and artificial intelligence, enhances corrosion monitoring, prediction, and decision-making, which are fundamental for a good future in the early detection of failures [84]. These approaches leverage large datasets to identify patterns, anomalies, and correlations that can improve corrosion management strategies. Advances in remote sensing technologies and real-time monitoring systems enable continuous and remote monitoring of corrosion parameters and provide valuable data for proactive corrosion management, allowing for early detection of corrosion, timely interventions, and improved asset integrity management.

The future of cellular automata in corrosion research will be about the integration of advanced computational techniques, such as machine-learning algorithms, and data-driven approaches. This integration allows for more accurate and predictive corrosion models by leveraging large datasets and optimizing model parameters based on experimental or real-time monitoring data [21]. In addition, the incorporation of electrochemical considerations in cellular automata models can better capture the fundamental mechanisms and behavior of corrosion, leading to more realistic simulations and predictions. Multi-scale modeling is a growing trend toward coupling cellular automata models with other modeling techniques, such as finite element analysis or computational fluid dynamics [59].

The integration of cellular automata in corrosion management generates models with corrosion monitoring and control systems that will enable real-time feedback and adaptive control strategies. This integration can enhance the effectiveness of corrosion prevention and mitigation measures by dynamically adjusting parameters based on evolving corrosion behavior. Real-time monitoring techniques in cellular automata models allow for dynamic modeling of corrosion processes. By integrating data from sensors and monitoring systems, researchers can obtain more accurate and up-to-date information on corrosion behavior, enabling proactive corrosion management strategies [90]. At the same time, the focus is on refining model parameters, algorithms, and simulation techniques to improve the accuracy and reliability of predictions, ultimately enabling more effective corrosion management strategies. Cellular Automata models have the potential to be used in the design and optimization of corrosion-resistant materials used for the storage of energy by molten salts [46,70]. Future perspectives in cellular automata models of molten salt corrosion involve the development of more sophisticated models that can accurately capture the complex corrosion mechanisms specific to molten salt environments. This includes considering factors such as electrochemical reactions, mass transport, and the interaction between molten salts and the material surface.

## 5. Conclusions

Cellular Automata offers a promising avenue for advancing corrosion management through the provision of a robust framework for modeling and simulation. Their effective capture of the intricate dynamics of corrosion processes presents an opportunity to develop predictive models, corrosion-resistant materials, and real-time monitoring systems. Although challenges remain to be addressed, the prospects of utilizing Cellular Automata in corrosion management appear encouraging. Continued research and development in this field have the potential to make significant advances in corrosion control and prevention. Most of the studies conclude that the Cellular Automata model has tremendous potential for simulating the corrosion process of different alloys under various conditions. It can generate complex models by employing simple rules, and the obtained results closely resemble experimental behavior. Furthermore, the model’s ability to represent multiple types of physical phenomena makes it a powerful tool that can help researchers view the studied phenomena from a different perspective.

## Figures and Tables

**Figure 1 materials-16-06051-f001:**
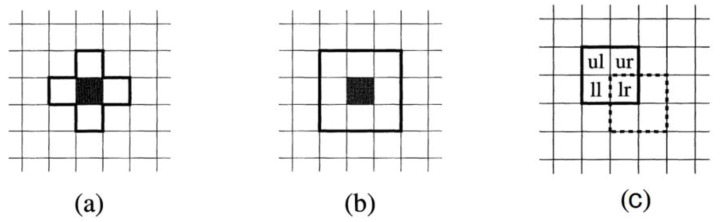
Most common neighbors. (**a**) Von Neumann. (**b**) Moore. (**c**) Margolus.

**Figure 2 materials-16-06051-f002:**
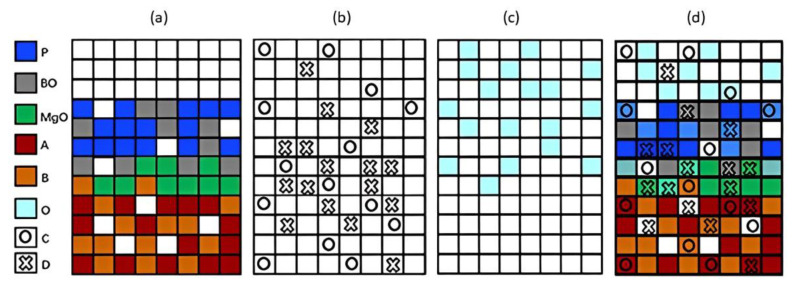
Schematics of the Cellular Automata model. Considering A = Cr; B = Ni; O = O_2_; C = Cl_2_; D = CrCl_4_; Mg = MgO; P = MgCr_2_O_4_; BO = NiO, respectively. (**a**) A grid of fixed sites. (**b**) A grid of mobile sites that diffuse towards the metal. (**c**) A grid of sites that can diffuse in the outer corrosion layer. (**d**) The general scheme of the grid in the Cellular Automata model [32].

**Figure 3 materials-16-06051-f003:**
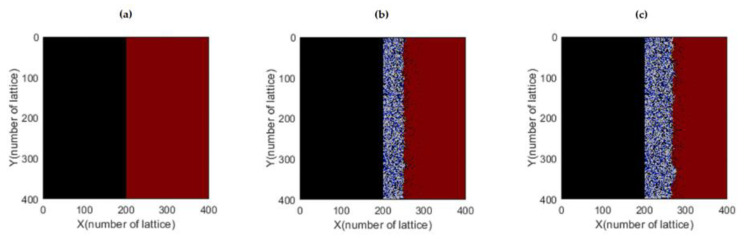
Simulation of corrosion layer growth at different iterations. (**a**) 40 iterations. (**b**) 25,000 iterations. (**c**) 36,000 iterations [32].

**Figure 4 materials-16-06051-f004:**
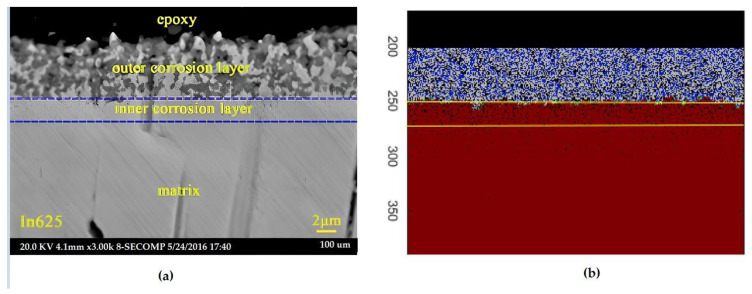
Comparison of simulation morphology and experiment SEM photograph: (**a**) cross–sectional SEM photograph of In625 after 21 days of immersion at 600 °C; (**b**) Snapshots of simulation results of corrosion morphology for time steps Nt = 25,000 [32].

**Figure 5 materials-16-06051-f005:**
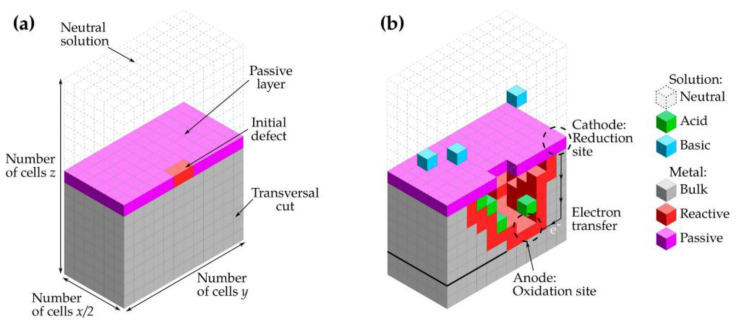
Transversal view schema of (**a**) the initial global matrix and (**b**) after the development of the pit [34].

**Figure 6 materials-16-06051-f006:**
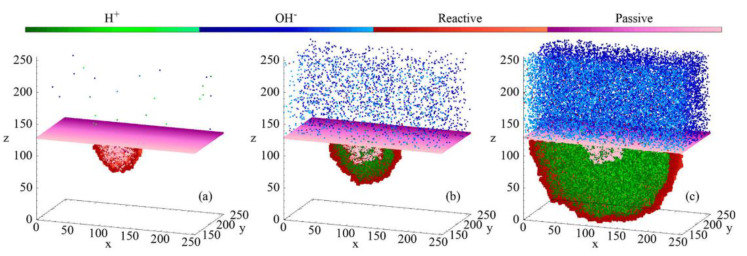
Side views illustrating different pitting corrosion regimes (Ndiff = 2000): (**a**) initiation (for t ≤ 44,400), (**b**) growth instability phase (approximately 500−time steps, around t = 44,600), (**c**) stable growth (from t = 45,000) [34].

**Figure 7 materials-16-06051-f007:**
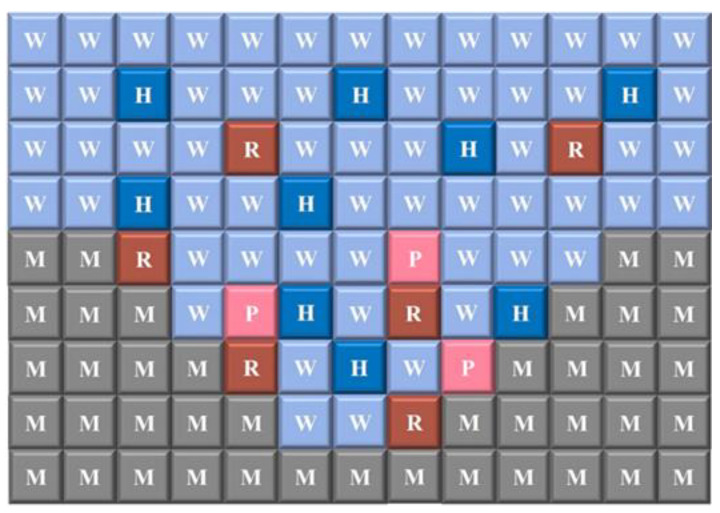
Schematic diagram of the cellular automata spatial model [59].

**Figure 8 materials-16-06051-f008:**
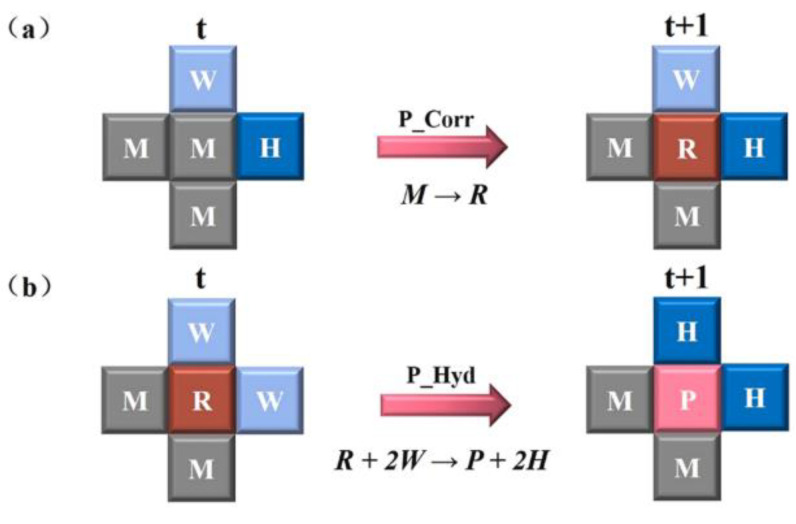
Cellular Automata corrosion evolution rules: (**a**) Oxidation of Fe to Fe^2+^ ions. (**b**) hydrolyzes Fe^2+^ ions to FeOH^+^ [59].

**Figure 9 materials-16-06051-f009:**
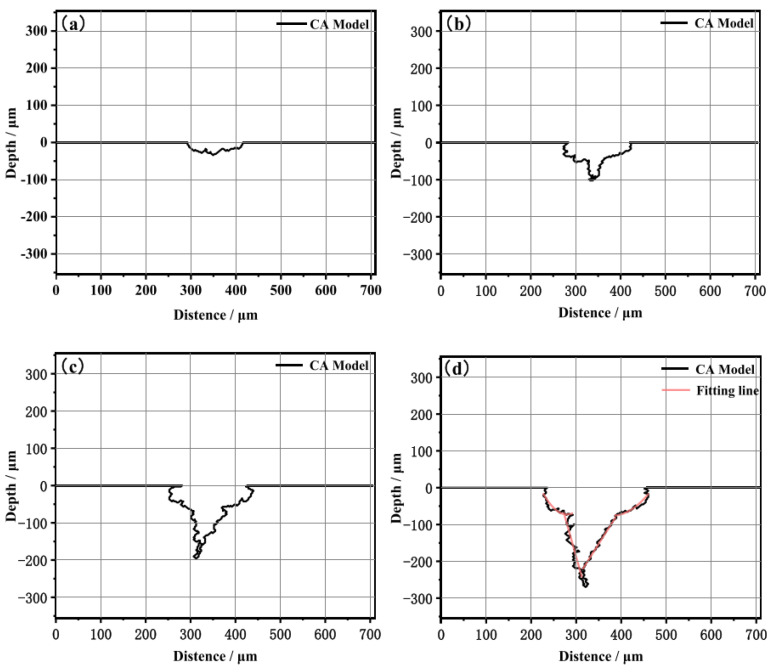
Evolution process of cross−sectional stress corrosion profiles: (**a**) 30 h, (**b**) 60 h, (**c**) 90 h, and (**d**) 120 h [59].

**Figure 10 materials-16-06051-f010:**
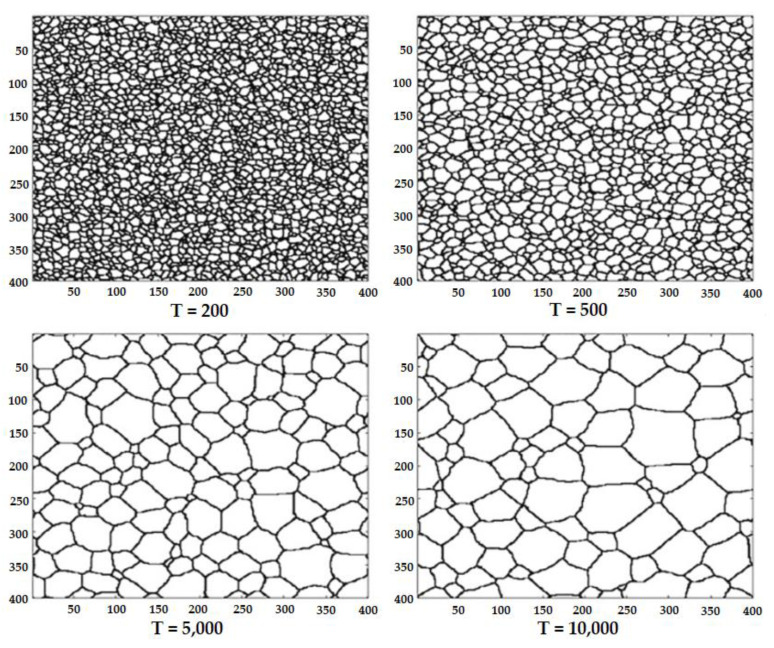
Grain growth process at different simulation time steps [46].

**Figure 11 materials-16-06051-f011:**

The simulation results of intergranular corrosion (T = 70,000) of the initial structure obtained at different simulation time steps are: (**a**) T = 500; (**b**) T = 1000; (**c**) T = 2000; (**d**) T = 5000 [46].

**Figure 12 materials-16-06051-f012:**
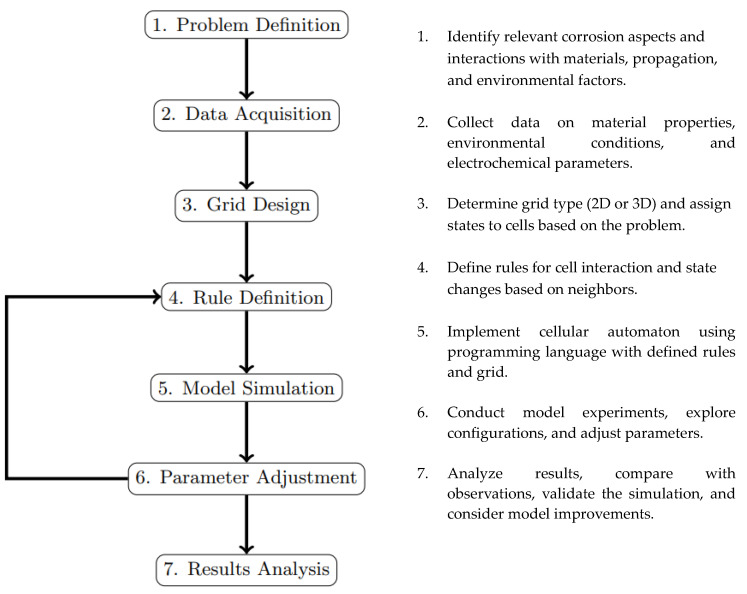
Methodology for implementing a cellular automata model.

## Data Availability

No new data were created or analyzed in this study. Data sharing is not applicable to this article.

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
