# Peer review of "Cellular Automata Modeling as a Tool in Corrosion Management"

_materials, 2023, doi:10.3390/ma16176051_

Round 1

Reviewer 1 Report

The topic authors chose is interesting and they provide valuable information regarding corrosion studies using cellular automata for corrosion processes simulations.

Page 2 lines 68-72: ” Additionally, it is essential to consider the applicable regulations and safety standards 68 from NACE [6] or another standard that can apply, pertaining to corrosion prevention 69 across various industrial sectors, including petrochemicals [7], aeronautics [8], shipbuild-70 ing and maritime infrastructure [9]–[12], energy [13], mining [14], nuclear waste [15], mi-71 crobial induced corrosion (MIC) [16], atmospheric corrosion [17], and among others.” I believe that “and” should be deleted.

Page 2 line 90 – page 3 line 91: “Cellular Automata models considering the impact of environmental factors, in the optimization of corrosion-resistant materials.” considering – consider?

Page 3 line 111: “(v) and CA can serve as educational tools to enhance” and (v) CA can serve as educational tool…

Page 3 line 120: “The AC model” or it should be CA model.

Page 3 line 125: “To clarify, for an entity to be classified as a CA (Counter Argument), it must adhere to the following structure.” CA stands for counter argument or cellular automata? If the same abbreviation is used for two different terms it is very confusing.

Page 4 line 142: “in the Moore neighborhood (b), there are six neighbors surrounding the central cell” six or eight neighbors – please check?

Page 5 line 207-208: “Wang et al [29], [44] investigated the corrosion of a metal in molten chloride salt and explored” which metal is studied it would be better to be more precise.

Page 5 lines 210-211: “Both experiments and simulations were conducted using the cellular automaton (CA) method.” rewrite to be clearer, it seems it could be understood as CA method was used for both experiments and simulations.

Page 5 equation 4 should be balanced.

Page 6 lines 220-222: “To establish the CA model, the chemical reactions involved in the corrosion process 220 were simplified by assigning letters to the compounds or elements present. A = Cr; B = Ni; O = O2; C = Cl2; D = CrCl4; Mg = MgO; P = MgCr2O4; BO = NiO.” Mg does not appear anywhere else only MgO.

Page 6 equation 6 – something is missing in the equation, maybe MgCl2 that is not included in further modeling, however the equation is incomplete.

Page 7 line 272. The authors focus on localized corrosion from this point forward so it should be presented as a new subsection 2.1.2. Localized corrosion model or similar.

Page 9 line 340: “SJ” the abbreviation should be defined as the others are.

Page 10 line 368: “In Figure 6, they showed the evolution of the pit. They indicated” please indicate who are they, refer to the authors of the paper.

Page 11 equation 19: equation is not balanced.

Page 11 lines 411-412: ” The evolution rules for corrosion in the CA model were formulated based on the previous equations” add equation numbers.

Page 11 line 413: “When a site M is in contact with at least one site H, reaction 1 will occur with” please check if number of the equation is correct.

Pages 13-14: equations 20-26 are not balanced.

Page 15 line 483: AC or CA model.

Page 15 lines 485-486: ”The development of intergranular corrosion in the studied steel by the researchers can be observed.” please rewrite the sentence to be clearer.

Page 15 caption of Figure 12: “cutomata” should be automata.

Table 1. Please be more precise and in the column Material add the exact metal that was used in the study. Add discussion and comments regarding the data presented in the Table. What can be concluded by the analysis of the summarized data?

Author Response

Dear reviewer. 

Thank you sincerely for your valuable comments, which have significantly enhanced the quality of our manuscript. We deeply appreciate and firmly believe that your insightful review significantly contributes to the advancement of knowledge in the realm of corrosion phenomena when applied to cellular automata models. Please see the attachment document regarding our response to your questions.

Best regards

Dr. Felipe M. Galleguillos Madrid.

Reviewer 2 Report

This paper was prepared and organized from the point that it showed ‘cellular automata modeling as tool in the corrosion management’. But some part of the manuscript needs to be revised as follow.

1. Please write Affiliations in English.

2. Write the titles of all figures and tables according to the regulations.

3. On line 93, please add references.

4. In the manuscript, please use cellular automata or CA uniformly.

5. In line 125, the abbreviation CA for counter argument can be confused with the abbreviation CA for cellular automata.

6. In the explanation of Figure 1(b), is moore's neighbors 6? It is identified as 8 in the picture.

7. Section 2.1.2 is missing. Please confirm.

8. All formulas should be written in consideration of stoichiometry. (Equations 4, 19, 20~26)

9. What is the AC model? (line 158, 403, 483)

10. In figure 8, match the expression in the figure with the expression in the text. (ex, P_Corr or Pcorr)

11. 4. Future Perspectives and Emerging Trends”, please check and add references per paragraph.

12. The first paragraph (line 591~594) of conclusion, it is shown as unconfirmed content in the text. Please add relevant content to the manuscript.

Author Response

(The authors gave the same response as above.)

Reviewer 3 Report

This review paper discusses the utilization of cellular automata as a valuable tool for evaluating, predicting, and controlling corrosion. The authors provide an overview of the fundamental principles of this approach and describe the primary models employed in the literature. This review article may be suitable for its publication in Materials after a minor revision.

Comments:

1. It is worth noting that finite element modeling (FEM), a widely used computational method in corrosion research with similar objectives as cellular automata, is not mentioned in the paper. We recommend including a discussion on the main differences between FEM and cellular automata and explaining why FEM was not considered for this review.

2. Regarding lines 32-33: "It has a significant economic impact, representing approximately 3.4% of the global Gross Domestic Product (GDP) [2]," this reviewer suggests providing a more recent value and including an up-to-date reference.

3. Please invite a native English speaker to proofread your manuscript, as there are several typos and grammatical errors. Here are a few examples:

Line 120, Line 158: The acronym "AC" is either undefined or should be corrected to "CA" (Cellular Automata).

Line 125: "To clarify, for an entity to be classified as a CA (Counter Argument)." Please avoid using the same acronym, as "CA" has been defined as Cellular Automata.

Figure 7 caption: Correct the word "diagramo" to "diagram."

4. Lastly, please ensure that the reference style adheres to the Journal guidelines provided at: https://www.mdpi.com/journal/materials/instructions

See above

Author Response

(The authors gave the same response as above.)

Reviewer 4 Report

The paper presents a review on Cellular Automata models for corrosion. The topic is interesting and the paper is overall well organized. However, some changes and clarifications are needed prior to the paper publication.

General comments:

- Figures such as 3 and 11 should have a legend. Alternatively, report the names of the components in the figure, as done in Fig. 4a.

Abstract

- "It discusses": The subjective of the sentence is not clear.

Uniform corrosion model

- "Fairén et al. [28] analyzed the evolution of surface roughness in the studied corroded metal": which metal?

Sections 2.1.2, 2.1.3 and 2.1.4

- The title of Section 2.1.2 is missing.

- Add a reference when you provide the definition of  corrosion types at the beginning of each section.

Section 3

- This section must be introduced more smoothly.

Conclusions

- Conclusions must contain a summary of what is reported in the manuscript, nothing new. Report in the previous section, what you say in Conclusions.

There are some typos. For example "Figure 7: Schematic diagramo f the cellular", "Chen et al. [56] developed a cellular automaton model".

Author Response

(The authors gave the same response as above.)
